# Suitability of Measurement Tools for Assessing the Prevalence of Child Domestic Work: A Rapid Systematic Review

**DOI:** 10.3390/ijerph18052357

**Published:** 2021-02-28

**Authors:** Nicola S. Pocock, Clara W. Chan, Cathy Zimmerman

**Affiliations:** 1Lumos Foundation, Peninsular House, 30-36 Monument Street, London EC3R 8NB, UK; 2Gender Violence & Health Centre, London School of Hygiene and Tropical Medicine, London WC1E 7HT, UK; cathy.zimmerman@lshtm.ac.uk; 3Independent Consultant, London WC1E 7HT, UK; clara.chan1@alumni.lshtm.ac.uk

**Keywords:** PRISMA, child domestic work, child labour, prevalence methods, critical appraisal, measurement properties

## Abstract

Child domestic work (CDW) is a hidden form of child labour. Globally, there were an estimated 17.2 million CDWs aged 5–17 in 2012, but there has been little critical analysis of methods and survey instruments used to capture prevalence of CDW. This rapid systematic review identified and critically reviewed the measurement tools used to estimate CDWs in Low- and Middle-Income Countries, following PRISMA guidelines (PROSPERO registration: CRD42019148702). Fourteen studies were included. In nationally representative surveys, CDW prevalence ranged from 17% among 13–24-year-old females in Haiti to 2% of children aged 10–17 in Brazil. Two good quality studies and one good quality measurement tool were identified. CDW prevalence was assessed using occupation-based methods (*n* = 9/14), household roster (*n* = 7) and industry methods (*n* = 4). Six studies combined approaches. Four studies included task-based questions; one study used this method to formally calculate prevalence. The task-based study estimated 30,000 more CDWs compared to other methods. CDWs are probably being undercounted, based on current standard measurement approaches. We recommend use of more sensitive, task-based methods for inclusion in household surveys. The cognitive and pilot testing of newly developed task-based questions is essential to ensure comprehension. In analyses, researchers should consider CDWs who may be disguised as distant or non-relatives.

## 1. Introduction

### 1.1. Rationale

Child domestic work is a largely invisible form of child labour, and a neglected type of domestic work compared to adult domestic work [1]. Child domestic work is a global phenomenon that affects the lives and futures of millions of children around the world. Globally, there were an estimated 17.2 million child domestic workers (CDWs) in 2012, of whom two thirds were girls and approximately 65% were aged 5–14. Over one-fifth of CDWs (3.7 million) were estimated to be in hazardous work [1]. Because of the relatively hidden nature of child domestic work, CDWs are particularly vulnerable to abuse and exploitation in private households, where they almost always fall outside the remit of national employment legislation. CDWs, especially those who are trafficked, are vulnerable to sexual, physical and psychological violence and coercion, which is often perpetrated by employers. Restrictions on their freedom of movement is common and it means that most CDWs have limited or no access to healthcare, education, other types of support services or job opportunities. Relative to children conducting household chores within their own home, live-in CDWs in particular face risks of abuse linked to blurred lines in the employing family, where they are neither seen as workers nor as family members [2]. CDW is often considered to be a form of ‘safe’ work by parents relative to other occupations, and/or can be an alternative to early marriage for girls [3,4].

Surveying and estimating prevalence of CDWs poses numerous challenges, which have hindered programming and policymaking to support them. First, there is no globally established definition of CDW, because CDW currently falls within various occupation terminology (e.g., domestic worker, housemaid, servant, housekeeper). Similarly, there is no globally agreed list of activities that are deemed to constitute domestic work [5]. As a result, Labour Force Surveys (LFS) often fail to fully capture children whose principal occupation is domestic work. CDW’s activities may not be considered employment, especially if no cash payment is involved, e.g., in-kind payments such as food, shelter or expenses. Consequently, CDWs may be reported as conducting unpaid household chores as a household member despite being, child domestic employees [5]. CDWs may be engaged in domestic work (DW) as a secondary job, which is not always captured in LFS or Child Labour Surveys (CLS). Trafficked CDWs may be deliberately concealed. Furthermore, the post-survey coding of occupation and the branch of economic activity that is used to establish occupation and industry in LFS is prone to errors (see Figure 1). It is not uncommon for CDWs in Low- and Middle-Income Countries (LMICS) to be misreported by employers as ‘fostered’ or adopted children doing unpaid household chores [5,6]. Consequently, CDWs are frequently missed in census, household, LFS and CLS. To develop accurate prevalence estimates of CDW, valid and reliable prevalence measurement tools are needed.

Figure 1 presents the statistical classification methods for CDWs in household surveys. The most commonly used methods are the ‘Household Roster’, ‘Occupation-based’ and ‘Industry-based’ methods. These methods are either standalone approaches or applied in combination with task-based methods, which is a newer way to enumerate workers. This review examines use of these different methods and the implications for accurate measurement of CDWs.

Current global CDW estimates rely on the industry approach, which identifies CDWs based on the activity or field that they work in, i.e., activities of households as employers of domestic personnel (Figure 1) [1]. Used alone, the industry approach may not accurately enumerate CDWs, who become subsumed in wider prevalence within other groups of workers in households, such as gardeners, security staff, cooks, babysitters, all of whom fall under ‘activities of households as employers of domestic personnel’ [5].

To consider potential improvements in the measurement of CDW, we conducted a rapid systematic review to identify and critically review measurement tools for the prevalence of CDWs in LMICs.

### 1.2. Objectives

The aim of this review was to identify and critically review measurement tools used to estimate prevalence among CDWs in LMICs. The objectives were to: (1) describe prevalence methods and questions; (2) critically review both study quality and measurement tool quality; (3) recommend promising question sets for CDW prevalence.

This review was conducted using an overarching protocol for four CDW-related reviews, which had wider PICOS criteria beyond CDW prevalence only (see Selection criteria below). In this review, the population included CDWs aged up to 18 and the outcome was CDW prevalence. Interventions and comparators were not applicable. All quantitative study types were eligible (cross sectional and longitudinal surveys, primary or secondary data analysis) as long as they included an element of random probability sampling.

Because of the potential measurement tool suitability in similar populations, we included studies designed specifically for use with CDW, as well as studies with Child Labourers/Working Children, Vulnerable Girls or Children (e.g., street children), where CDWs were reportedly included.

In this review, we define CDW as a person under 18 years old performing domestic chores and caring tasks in the *home of a third party*, with or without remuneration. We make a distinction from children performing household chores or providing care within their own homes with immediate family. CDWs may be living and working with distant or close relatives (e.g., aunts and uncles). They may also be working under “fostering” type arrangements in the home of a third party. Children may be living with such employers or relatives (live-in CDW), or they may be living elsewhere and commuting to the household of work (live-out CDW). The types of tasks performed by CDWs appears to be similar across countries. For example, tasks include care of the children and the elderly, fetching water and wood, tending to animals, cleaning, cooking and purchasing daily household essentials. In most countries, CDWs are primarily, but not exclusively, girls [3,7].

## 2. Methods

### 2.1. Search Strategy

The protocol was registered in the PROSPERO database of systematic reviews, registration number CRD42019148702 [8]. The protocol search strategy and search terms were designed to cover four scoping and rapid systematic reviews for the population of interest (CDWs and employers), including the current review on CDW prevalence. Search concepts and terms were developed by review leads with the London School of Hygiene and Tropical Medicine library team.

A set of overarching databases were searched by an LSHTM librarian (see Appendix A), which included MEDLINE, EMBASE, Global Health, Econlit, Web of Science, International Bibliography of the Social Sciences from inception to 4 June 2019. Two reviewers (CC and NP) searched for further citations from grey literature including UN agency websites (see Appendix A). The studies were screened against inclusion and exclusion criteria.

### 2.2. Selection Criteria

During the title and abstract screening stage of the four CDW-related reviews, studies were eligible for inclusion if they included: (1) CDWs (up to 18 years old) and young adult domestic workers (18–25 years old) or employers of CDWs; (2) either occupational outcomes, health, education outcomes, risks or abuses, CDW prevalence, economic outcomes or outcomes related to employer attitudes or behaviour. Observational studies and intervention evaluations reporting on relevant outcomes for each review were eligible. Studies could be observational studies (cohort, case-control, cross sectional/post only assessments), qualitative studies, quasi experimental and experimental studies. Systematic reviews were used for the purpose of backwards/forwards citation tracking. Studies from LMICs were eligible, in addition to the following High-Income Countries (HIC) where migrant domestic work from Southeast Asian countries is common: Singapore, Taiwan, Macau, Hong Kong, Brunei. We selected these countries because Southeast Asia is the focus of a larger body of work on CDW for which this review was a component. Date limits of 1990–2019 were applied, as studies pre-1990 were not relevant and to be sure we accounted for recent evidence. Only studies in English were eligible.

Studies were excluded if they: (1) focussed on adult domestic workers only (aged 25 or above); (2) included only children or young adults (up to 25 years old) performing household chores or care work in their own homes with immediate family; and (3) featured CDW profiles, without reference to any of the relevant outcomes. Letters, commentaries, conference abstracts, books and book reviews were excluded.

At the full-text screening stage for this review on prevalence, inclusion criteria were narrowed to include: (1) studies specifically on Child Labour, CDW, vulnerable girls or children (e.g., kin care, street kids) which included crude numbers for CDWs within the sample and/or appropriate denominators to assess prevalence within a wider reference population; (2) an element of random probability sampling was used in the study.

Exclusion criteria were narrowed further: (1) No denominator was reported for the overall study; (2) no details were given on the design of measurement tools used to assess the relevant outcomes. Due to time constraints, we did not conduct citation tracking of included studies.

### 2.3. Data Extraction

Two reviewers (CC and NP) screened downloaded titles and abstracts for potential inclusion. The same reviewers then assessed the full text of potentially eligible papers against the overarching inclusion criteria.

At the title and abstract screening stage, citations were divided in half between two reviewers (CC and NP) with each screening titles and abstracts against inclusion criteria in the protocol and screening form (see Appendix A). The same two reviewers reviewed each other’s list of studies to potentially include (‘Maybes’), for agreement. The final list of ‘Maybes’ at title and abstract stage were reviewed again by the same two reviewers, to check consistency of applying the inclusion criteria. In addition, the reviewers checked a random 10% of each other’s respective excluded and included studies at the title abstract stage, to ensure consistency. Full-text screening was conducted independently by the same two reviewers.

A data extraction form was developed and piloted by CC. Data from 75% of included papers were extracted by CC while NP extracted data for 25% of the studies; NP independently extracted data from a random sample of 10% of CC’s included studies as a check and vice versa; disagreements were resolved by discussion.

Data were extracted on study design, study population and sample characteristics, sampling method and prevalence, violence and health outcomes and ethical considerations. Sample characteristics included whether participants were CDW as defined by participants, providers or researchers: no restrictions were placed on the method by which CDW status was assessed. To assess measurement tools, we extracted data on: method of assessing the CDW prevalence; information on validity, reliability of measures and any translation of the survey instrument; modifications for cultural sensitivity to questions, and; method of survey administration [9].

### 2.4. Data Analysis

In this review, we focus primarily on measurement properties of instruments used to detect CDWs, and actual questions used, where available. We do not comment on statistical methods used for prevalence estimation. Pooled estimates were not calculated due to heterogeneity in study sample selection, definitions and methods of assessing CDW prevalence. We focus on narrative synthesis and describing the study results, limitations and implications for accurate CDW prevalence measurement.

### 2.5. Critical Appraisal

#### 2.5.1. Overall Study Appraisal

The methodological quality of studies was appraised by two reviewers; CC appraised 75% of studies while NP appraised the remaining 25% of studies (the same studies for which data were extracted). Both reviewers then independently appraised 50% of each other’s allocation to ensure consistency in applying the quality criteria, and no significant disagreements were found.

We used the Joanna Briggs critical appraisal tools (CAT) according to study design for cross-sectional prevalence studies. The JBI prevalence CAT included nine questions on sampling methods, method of measuring the outcome, analysis methods and response rate. We used the following classification system to indicate overall quality of studies: 0–50% Poor, 51–75% Moderate, 76–100% Good. Low critical appraisal scores were not used to exclude studies but are referred to in the discussion.

#### 2.5.2. Measurement Tools Appraisal

We adapted a quality appraisal tool (QAT) from a previous systematic review examining suitability of mental health measurement tools for trafficked persons [9]. The QAT consisted of six questions across the following categories: Validity (pilot-tested); Validity (used in similar population previously); Reliability (internal consistency tests, e.g., Cronbach’s alpha, were conducted); Reliability (inter-rater reliability, including appropriate training of data collectors); Method of tool administration (appropriateness of surveyors with specific information on who they were) and a question on Cultural adaptation (back-translation; modifications for cultural appropriateness). All questions were binary except for “Cultural adaptation” where a maximum score of 2 could be attained. The total maximum score was seven. Measures appraised between 0–3 were considered “poor” quality, measures scoring 4–5 “moderate”, and measures scoring 6–7 “good” quality. Where studies gave no information on a particular domain, 0 was assigned.

## 3. Results

Academic database searches and purposive grey literature searches returned a total of 6573 unique records after removing duplicates (Figure 2). A total of 211 papers were screened at the full-text stage, with 14 papers finally eligible for inclusion.

### 3.1. Characteristics of Included Papers

All included studies were cross-sectional designs (Table 1). The sampling for six studies included household surveys based on census data or government statistics office population projections [10,11,12,13,14,15]. Four studies conducted household surveys based on sampling from new baseline household listings [16,17,18,19]. One study was based directly on census data for several countries [20] and one study used an Injury Surveillance System (ISS) to capture CDW [21]. The studies were mainly conducted in Ethiopia [14,15,17], Haiti [11,13] and Bangladesh [10,21], followed by Brazil [22], South Africa [18], Cambodia [19], Vietnam [16], Indonesia [12]. Two studies included multi-country samples [7,20]. 

Overall, two studies were rated to be of good quality [11,17]. Five studies were of moderate quality [10,12,13,19,21], while seven were poor quality [7,14,15,16,18,20,22]. Importantly, the identification of child domestic work was unclear in all studies. This outcome gap was likely due to the absence of consistent or standardized CDW definitions and use of unvalidated self-reporting methods, which score poorly according to JBI critical appraisal criteria. Studies that scored poorly were usually secondary data analyses, which offered limited information on sample selection, response rates and statistical analysis methods. Critical appraisal scores at the measurement tool level and study level are detailed in Appendix A.

The critical appraisal ratings of the measurement tools differed from overall study quality in most cases (Appendix A). Only one prevalence measurement tool achieved a rating of good quality [10], while three tools were moderate quality [12,15,19] and the rest were poor quality. Those that were rated poorly offered scarce or no information on measurement validity or reliability. Limited information was presented on how final tools were developed in the vast majority of studies. For example, just four studies explicitly mentioned pilot testing instruments [10,11,15,19]. Only five studies mentioned cultural adaptations of the instrument to local contexts [10,11,12,15,17]. Slightly more information (six studies) was cited on training of enumerators to collect prevalence information [10,12,13,15,17,19]. Further detail on information used to score prevalence measurement tools is in Appendix A. Across fourteen studies, nine did not report any ethics or informed consent processes, although this was not considered as part of the measurement quality.

### 3.2. CDW Prevalence Estimates

Prevalence estimates varied widely, and the lack of information on denominators means that some studies only provided weighted estimates for geographic areas rather than as a proportion of all children by age groups, or proportion of CDW among all child labourers or all domestic workers (Table 1). In nationally representative surveys, 17% of females and 12% of males aged 13–24 reported ever being a CDW in Haiti [11], while 2% of children aged 10–17 in Brazil and 4% in Paraguay were estimated to be CDWs [7,22]. Ten percent of children aged 7–17 were estimated to be CDW in Phnom Penh [19]. Of studies examining CDW as a proportion of working children, estimates varied by sampling method. Two percent of working children (62% of whom were boys) were estimated to be CDW in a South Africa household survey [18], while 54% of working children (all girls) captured in an ISS were CDW in Bangladesh [21]. In a small rural Ethiopian town, 77% of working children were CDWs [15].

### 3.3. Methods and Questions Used to Determine CDW Prevalence

Prevalence was measured in different ways across the various studies (Table 2. Further detail in Appendix A), Some studies explicitly used multiple methods in order to compare prevalence measurement [7,12,20]. Most studies used occupation-based methods to enumerate CDWs (*n* = 9), which requires collection of detailed occupational data (see Figure 1). Seven studies relied on the household roster, by asking household heads whether anyone in the household was a domestic worker. The injury surveillance study relied solely on the household roster reporting method alone [21]. Four studies used industry-based questions reliant on specifying the main activity or field of work (see Figure 1). Four studies included specific task-based questions, which usually include binary Y/N questions to a list of household tasks (Figure 1), however, task-based methods were only used to formally enumerate CDW prevalence in one study, which included a follow-up question on whether wages or payment in-kind was received for conducting the tasks [12]. Six studies used a combination of prevalence methods. 

Studies that used household roster reporting alone were reliant on household heads declaring that they had a ‘domestic worker’. These studies usually found lower estimates compared to occupation-based methods. Using microsamples of census data from six Latin American countries, Levison and Langer (2010) produced ‘best guess’ estimates using household roster and occupation-based methods and included CDWs who declared they were unemployed during the past month. Prevalence determination through household rosters alone was very low for most Latin American countries, ranging from 11–39% [20]. In a sensitivity analysis, 25% of potential live-in CDW declared as ‘other relatives’ or ‘non-relatives’ in relationship to household head were counted to assess how much this ‘Cinderella’ or ‘disguised’ CDW group would augment estimates. In Brazil, using this method resulted in an addition of over 150,000 CDWs aged 10–17 [20]. Overall, relying solely on occupational information resulted in 93–100% of CDWs being identified across all six countries included in the study. Occupational information was a more reliable way of identifying CDWs compared to relying on the household roster question alone.

Similarly, Lyon and Valdivia (2010) assessed disguised CDWs by including non-relatives, other relatives and nephews/nieces alongside occupational information on hours spent doing unpaid household work. By including disguised CDW in Paraguay, this increased estimates by over one-third (from 2.5% to 4%), even when applying a high threshold of working 35 h or more per week spent on housework [7]. Among CDWs in Paraguay, approximately 31% were disguised, while 56% were commuting (live-out) CDW. In Uganda, including ‘disguised’ CDW increased estimates from approximately 1.5% to 5.6%, when applying a threshold of 28 h a week or more spent on housework [7].

Despite known limitations, use of the household roster reporting question alone (‘domestic worker’) enumerated many CDWs in the Colombia census and the Bangladesh Injury Surveillance System (ISS) studies. In Colombian census data, 80% of CDW could be identified using just the household head relationship [20]. There appeared to be a high declaration of CDW in the ISS household survey in Bangladesh, where over half (54%) of child labourers were CDW [21].

One moderate quality study (with moderate quality measurement tool) used a task-based ‘probing’ method to compare DW prevalence to household roster, occupation and industry-based approaches [12]. In Indonesia, a household survey was conducted based on the Indonesian Labour Force Survey format. The study team added the task-based probe at the end, from which nationally representative estimates were calculated. The study combined a task-based list followed immediately by a question about whether payment (in cash or in kind) was received for conducting any of the above-mentioned tasks. Crucially, this question about payment helped distinguish CDW from children doing unpaid household chores. Overall, more DWs were picked up using the task-based probe compared to the other approaches. Combining DW prevalence using all methods, the authors calculated a crude coverage ratio of 1.51 and revised estimates based on an adjusted coverage ratio by strata (rural, urban, region, district type). While there were 54,977 CDWs aged 10–17 in unadjusted estimates, this figure increased by 55% to 85,574 when applying the adjusted coverage ratio [12]. This study asked questions of household heads only. Enumerators were all graduates with training in public health (Appendix A. This was the only survey to combine methods in this way.

The highest scoring study (moderate study quality, good quality measurement tool) included task-based questions with both household heads and CDWs themselves in Bangladesh [10] (Appendix A). However, it was not clear which methods were actually used to determine prevalence. Nationally representative estimates for CDWs (*n* = 421,486) were calculated based on a household survey. The Bangladesh study provided detailed information on enumerator training which contributed to its high score, with training lasting a week and including mock interviews and field practice. Only enumerators with satisfactory performance during training went on to be recruited for the study (Appendix A).

Two studies, with poor and moderate overall and tool quality respectively, used enumerators from National Statistics Offices in South Africa and Cambodia [18,19]. The moderate quality Cambodia study did not use task-based methods to enumerate CDW prevalence, but included task questions in the survey for both household heads and CDW directly [19]. CDW prevalence was based on occupation and industry-based methods.

### 3.4. Live-In versus Live-Out CDW Prevalence Determination

Three studies included estimates for live-out CDW [7,12,20]. The methods used in the one primary study for live-out CDW conducted in Indonesia, included a task-based module that was repeated for ‘jobs for this household’ (live in) and ‘jobs for other household’ (live out) [12]. Secondary studies used occupation based data, so that live-out CDWs were enumerated in their own families but identified by their occupation information as CDW [7,20].

Some studies included questions to determine live-out status but did not explicitly include live-out estimates. Methods for identifying the child’s residence status included asking CDWs directly whether they lived with a parent [17]. One moderate quality study (with a poor quality measurement tool) included an extensive migration module with both household heads and with CDW directly, which allowed for extensive sub-group findings by migration and orphan status [13].

One primary study discussed difficulties of sampling live-in CDWs [16]. The authors explained that local authorities were reluctant to allow survey teams to conduct baseline household listings as having CDW (as a form of Child Labour) because this would give the district a poor image. In addition, the majority of CDWs were not registered with the local police as temporarily resident in employer’s households. The survey team conducted follow up visits to randomly selected households to ensure the reliability of information provided by household heads in a household roster. Employers were generally reluctant to declare they had CDW, with many employers stating that CDWs were relatives in local authority records [16].

## 4. Discussion

### Key Findings and Recommendations

As our review findings demonstrate, there are currently very few studies measuring the prevalence of child domestic work. Many children in domestic work are likely to remain hidden based on most commonly used methods, which rely on household head, industry and occupational approaches (Figure 1). As household head, occupation and industry-based questions are included in standard labour force surveys and child labour surveys, we recommend that an additional task-based module be included in household surveys to improve estimates. Nieces/nephews, non-relatives and other relations should be included as potential CDWs from the household roster, when combined with occupation, industry and or task-list data (Figure 3). These recommendations are shaded in green in Figure 3.

The critical assessment of 14 prevalence measurement tools conducted as part of this review indicates that the most frequently used tools are those that use occupation-based questions and questions based on the child’s relationship to the household head in rosters. However, our findings suggest that the four studies using a task-based list of questions seemed most promising to achieve more accurate estimates. A list of specific tasks is more likely to be objectively understood and answered by respondents. Task-based questions appear to offer a major advantage over other methods because they do not depend on the household heads or CDWs themselves to determine whether domestic work is ‘work’ or ‘employment’. Instead, they simply identify who is doing each specific task. As some child labour studies have shown, children do not necessarily perceive their agricultural or domestic tasks as ‘work’ [23,24], implying that when surveys limit question phrasing to words such as ‘work’, ‘job’, ‘employment’, or ‘industry’, this may lead to undercounting of CDWs. 

This type of task-based list is similar to methods applied in violence studies, which use ‘act-based’ questions, such as the conflict tactics scale, which asks women about specific acts of violence (e.g., slap, hit with object, kicked) [25,26] rather than broad questions such as ‘were you physically abused?’. Questions that rely on interpretations of concepts such as abuse, violence, work or exploitation have proven to elicit highly subjective responses that depend on how respondents understand ‘abuse’ or ‘work’ in their own lives and in their cultural context. Similar to asking about specific acts of violence, a task-based question set enables the respondent to consider whether and how much they do that specific task (e.g., sweeping, washing dishes, etc). And, at the same time, like surveys on violence, tasks can be very context-specific, so question sets will have to be developed based on the results of cognitive interviews to identify locally relevant tasks. Cognitive interviews are a useful method to detect how respondents themselves understand the concepts and interpret each question [27].

Task-based methods can be used in two main ways. As suggested by Suhaimi & Farid (2018), their task-based approach combines a lower cost, smaller household survey with larger labour force surveys. In this way, researchers will calculate adjustment factors based on task-based data, which will be applied to existing labour force data [12]. Alternatively, a task-based module can be integrated into future labour force surveys to generate more reliable estimates. The costs in terms of time to administer this module are likely to be small, relative to gains in accuracy for CDW prevalence estimation.

Our findings also highlight the perils of relying solely on surveys that query the relationship to household heads to detect child domestic work, particularly because of the stigma associated with employing children as domestic workers in many countries. This household roster method is probably only suitable in countries where CDW is highly acceptable, perhaps in studies like the *Injury Surveillance Study* where enumerators conducted longitudinal household surveys, which may have enabled them to build rapport and trust with household heads [21]. In contrast, census surveys, labour force and child labour surveys are infrequent in LMICS, and are often administered by National Statistics Office enumerators, which makes building a rapport very unlikely. 

As part of this review, we briefly examined other well-known, publicly available surveys conducted with or about children, to assess how frequently CDW prevalence questions were included. The *UNICEF Multiple Indicator Cluster Survey* (MICS) instrument includes a household relationship question, as do the *Demographic and Health Surveys*. Used alone, this method is unlikely to yield accurate CDW estimates given the reliance on household heads to declare a child in their household is a ‘domestic worker’. UNICEF MICS has a child labour module that includes responses about household tasks and hours spent on tasks from household head [28]. Using this type of approach, it might be possible to examine disguised CDWs using other categories in the household roster (e.g., which includes foster/adopted/stepchild). Elsewhere, the *Young Lives* surveys include questions on how children use their time, which are administered directly with children, but these surveys do not include explicit CDW methods or questions [29]. Overall, without substantial probing, these surveys are unlikely to fully detect CDW.

Beyond the questions recommended in Figure 3, accurate reporting of working hours is another important element of CDW prevalence estimation, particularly to identify hazardous CDW. Emerging child labour measurement studies highlight the importance of considering who is the respondent for questions such as these. Household head and adult proxy respondents may underreport the hours that household members spend doing household chores, especially tasks done by the girls in the household [23]. Elsewhere, children themselves were more likely to report hazardous exposures, such as using fire, gas or flames, compared with reporting by adult proxies [30]. Child friendly time-use methods that allowed children to express tasks in their own words yielded more detailed task information than when tasks were reported by adults [24]. Overall, task-based modules with household heads and with children themselves, preceded by cognitive interviewing and pilot testing can yield stronger question sets. Additionally, to improve accuracy, some child labour surveys have included a component that assesses the attitudes of the household head, which could be controlled for in statistical analyses when examining differences in children’s and adults’ responses to CDW prevalence questions. 

When considering study quality, in some cases, study quality was good, but measurement tool appraisal was low [11,17]. That is, while studies on CDW may be well conceived in study design, sampling methods and analysis, development and piloting of instruments used to capture prevalence, health outcomes or risks is usually not well reported. We recommend that researchers write up these findings for wider learning in the child labour measurement community. 

Despite the various weaknesses, studies included in this review hint at the burden of adverse working conditions, violence and poor health outcomes associated with CDW. Over 99% of CDWs in Bangladesh worked seven days a week, nine hours a day on average [10]. Similarly, one study in Haiti which used a validated violence measure, indicated that over three-quarters (77%) of former female CDWs had ever experienced physical violence before the age of 18, and nearly half (40%) reported experiencing sexual violence [11]. Additionally, occupational safety risks to Haitian CDWs were clear based on data from currently working CDWs, which indicated that over 90% were using hot stoves or open fires or using sharp objects, with almost 80% using household chemicals [13]. In other studies using validated health outcomes measures, the prevalence of musculoskeletal pain was 17% higher among CDWs than children working in other sectors, with over half of CDWs reporting back pain linked to heavy, repetitive and monotonous work and awkward postures [31]. The psychological implications of child labour are also important. For example, among CDWs in Ethiopia, 5% had symptoms associated with depression, with 39% reporting difficulties performing everyday tasks as a result of depressive symptoms [32]. Similar findings are echoed in studies with adult domestic worker populations [33], but for child and adolescent girls these conditions can be especially pernicious given their developmental stage. During childhood and adolescence, biological shifts occur, which means youth require more sleep, and CDWs are often deprived of adequate sleep because of their long working hours [34]. Moreover, while youth are growing, toxins are more easily absorbed via children’s thinner skin, and their ability to excrete toxins is lower relative to adults [35]. Consequently, chemical exposure during childhood is especially dangerous because it can lead to neuro behavioural problems and reduced cognitive function, according to a recent systematic review of child work [35]. 

While there has been an understandable focus on risks associated with CDW, we know little about what constitutes beneficial or protective forms of child work. Selected domestic work tasks may be less harmful and even preferred by children, such as sweeping versus farm work [36]. Harm incurred by work depends on the types of tasks, duration and frequency at which children are undertaking them, as well as the conditions in which CDW are employed, which may include employer abuse or restricted movement [37,38]. Detailed task information is needed to ascertain hazard levels for particular tasks within different sectors is lacking in many studies on child labour and health and wellbeing, including for CDW. Until we combine this granular information with employment conditions and age-specific health impacts of work, it is impossible to ascertain which forms of child labour, including CDW, are empirically ‘hazardous’. 

While describing household characteristics was not an explicit objective of this review, some studies included features of the employing household. For example, Lyon & Valdivia (2010) found that disguised CDWs in Paraguay tended to reside in mid to lower income employer households, but in wealthier households, heads of household were more likely to report the presence of CDWs [7]. Live-in CDW tended to be disguised as non or other relatives, often rural-urban migrants, and more frequently were orphaned or living away from parents [7,13]. Because of the challenges associated with disguised CDWs, who are often reported as nieces or nephews, other relatives or fostered children, task-based modules appear to lead to better identification of CDWs, especially in low- and middle-income employer households. We recommend using all four question types in Figure 3, to ensure that CDWs are sufficiently captured. Using Suhaimi & Farid’s (2018) estimations in Indonesia, up to 50% of CDWs may be missing using standard household roster, industry and occupation-based questions alone [12].

Developing a single CDW prevalence module will require detailed examination of questions in existing tools (Appendix A), followed by pilot testing, cognitive interviews and adaptation in different contexts. The authors are currently working on a module for use in a planned CDW prevalence study in Myanmar, given anticipated serious undercounting of CDWs in current estimates [39].

## 5. Strengths and Limitations

We used a rigorous design and application of an adapted measurement critical appraisal tool to an emerging field of child labour measurement [23,30,40,41,42]. We identify and recommend questions that can be used in future CDW prevalence studies.

This review has some limitations. We opted to conduct a rapid systematic review because of its suitability for an emerging research domain and broad PICO criteria, while maintaining a rigorous approach to evidence synthesis. Reporting guidelines for rapid reviews (PRISMA-RR) are currently being developed [43]. We mitigated bias by searching multiple sources. We used PRISMA guidelines to report findings and note departures from full systematic review methods. Owing to resource constraints, we were unable to conduct full double title abstract screening by two reviewers. Use of a detailed screening flowchart (Appendix A), a 10% random check of each reviewers respective excluded and included studies, and frequent conversations during screening to check consistency of applying the inclusion criteria, hopefully mitigated against bias at this stage. 

As with all quality appraisals, scoring is reliant on study authors reporting adequate information on how studies were designed, and how tools constructed and tested. Secondary studies in particular typically do not include detailed information on how survey instruments were constructed in the original survey, which meant these studies scored poorly in this review. In many cases it is hence unclear whether a study was badly designed, or if it is merely badly reported. Authors should use guidelines and checklists such as Strengthening the Reporting of Observational studies in Epidemiology (STROBE) (observational studies) [44] and Consolidated Standards for Reporting Trials (CONSORT) (RCTs) [45] to ensure complete reporting of study findings, alongside detailed information on how measurement tools were constructed. 

## 6. Conclusions

Child domestic workers are among the hardest to reach and least visible child labour population. Our review findings suggest that the current methods have led to under-reporting of global prevalence of CDWs. Using more detailed task-based methods and including carefully crafted and tested approaches to detect CDWs who are potentially disguised as distant or non-relatives, will likely result in larger and more accurate prevalence estimates for CDWs. Given the potential very high prevalence of CDWs in LMICS, improving measurement should be a top priority for international agencies and donors concerned with child labour. We urge researchers and agencies leading larger household survey programmes as well as organisations and governments concerned with child labour to revise their approaches to include stronger question set modules in future surveys. Forgotten for far too long, child domestic workers deserve much greater policy and programmatic attention, and this starts with improved prevalence estimation methods.

## Figures and Tables

**Figure 1 ijerph-18-02357-f001:**
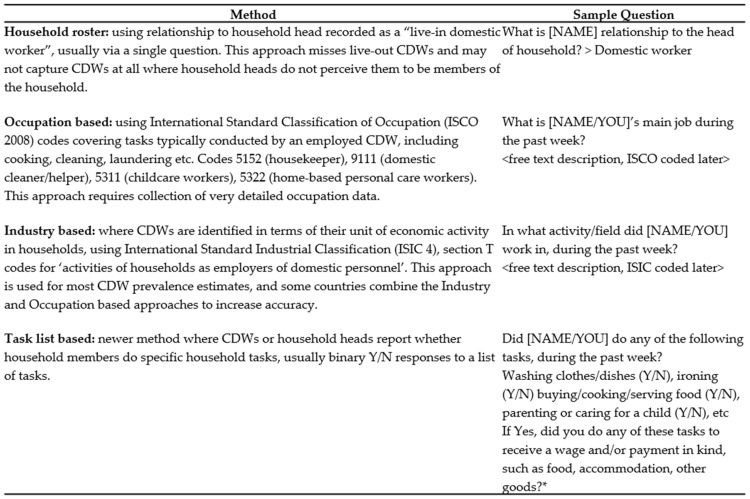
Statistical classification methods for child domestic workers (CDWs) in household surveys. Source: 2017 ILO Practical Guide to Ending Child Labour and Protecting Young Workers in Domestic Work, 2019 Pocock and Zimmerman Child Domestic Worker prevalence in Myanmar and Southeast Asia: Briefing note. * Example question conceived by authors.

**Figure 2 ijerph-18-02357-f002:**
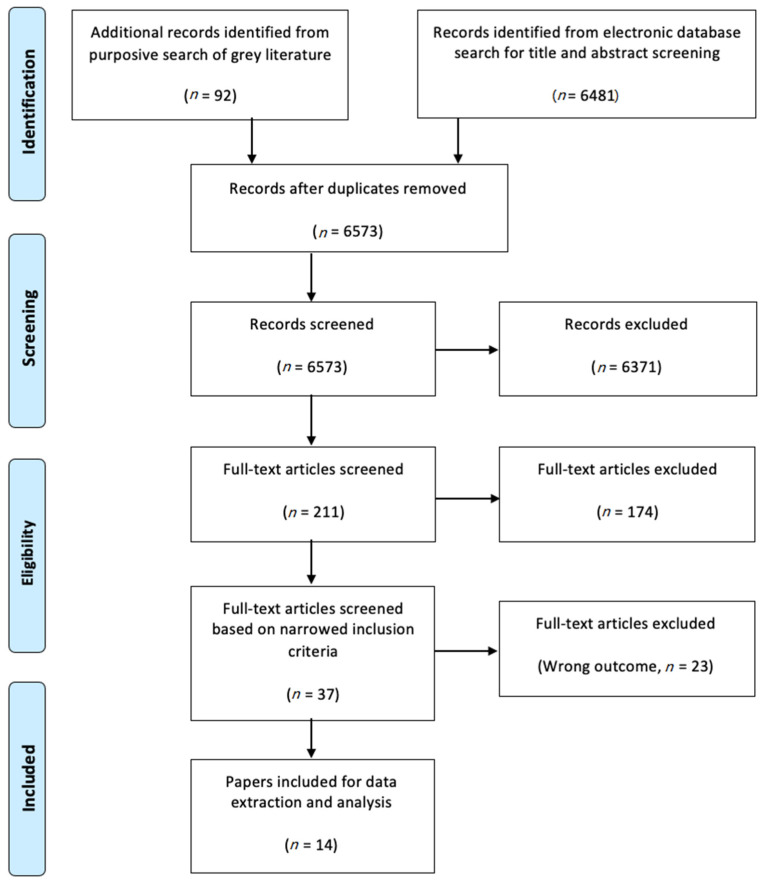
Flowchart of CDW prevalence study selection.

**Figure 3 ijerph-18-02357-f003:**
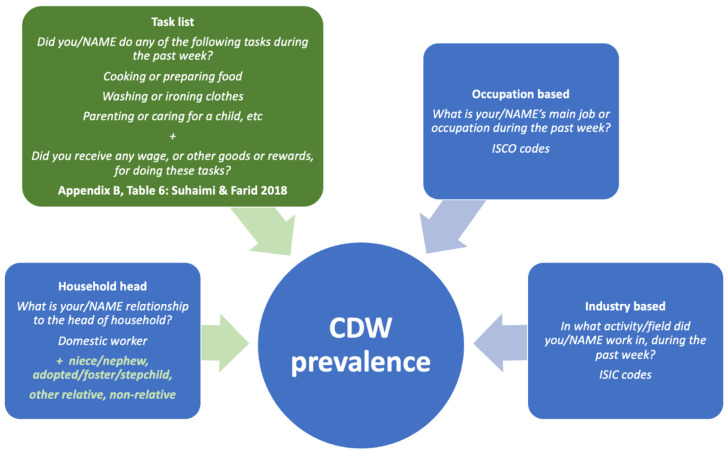
Recommended methods and questions for CDW prevalence in labour force surveys and child labour surveys.

**Table 1 ijerph-18-02357-t001:** CDW cross-sectional prevalence study characteristics (*n* = 14).

	Study	Setting	Sample Size	Measurement Tool and Sampling Method	Primary Outcomes	Prevalence Estimate for CDW	Study Quality	Tool Quality
1	Kedir & Rodgers 2018 * [14]	Ethiopian urban population	Domestic workers aged >= 10 N = 1500 households (12,000 individuals)	**Measurement tool:** Ethiopian Urban Household Survey, longitudinal 1994–2004 **Sampling method:** 1500 households in seven urban centres of Ethiopia (proportional sampling by urban population size, with systematic sampling of households at the ‘kebele’ level in each district	Prevalence estimates of domestic workers	**CDW aged 10 to 15:** 1994: 17.9% (93/520) 2004: 8.3% (43/517) **Denominator:** DW who specified age ^	Poor	Poor
2	Gilbert et al. 2018 * [11]	Haitian households displaced by 2010 earthquake (including IDP camps)	CDW (ever been restaveks) aged 13–24 (N = 451/2916)	**Measurement tool:** Nationally representative cross-sectional household survey of children and young people (Violence Against Children Survey 2012) **Sampling method:** Stratified, three-stage cluster design used to sample households and camps affected by 2010 earthquake, based on updated estimates from 2003 Haitian census	Prevalence of violence before 18 (physical, emotional, sexual) amongst CDW	**Ever restavek before age 18:** Female crude *n* = 281, weighted *n* = 225,989, (17.4%) Male crude *n* = 170, weighted *n* = 159,384 (12.2%) **Denominator:** 13–24y/o in sample reporting age when became restavek	Good	Poor
3	Dalal et al. 2016 * [21]	Child labourers in 3 rural sub-districts, Bangladesh	Child labourers aged 6–17 (N = 42 487, including N = 23,087 CDW)	**Measurement tool:** District-level injury Surveillance System (2006–2010)–baseline census, followed by periodic representative household survey in three selected sub-districts **Sampling method:** Child labourers selected from data captured by surveillance system	Prevalence of injury resulting in death or morbidity amongst child labourers across sectors (including domestic work)	**CDW:** 54.3% (*n* = 23,087/42,478), all of whom were female **Denominator:** child labourers 6–17 in sample	Moderate	Poor
4	Degraff et al. 2016 * [22]	Children in hazardous labour, Brazil	Children aged 10–17 (N = 60,678, including N = 1129 CDW)	**Measurement tool:** Nationally representative household survey (PNAD 2001) **Sampling method:** Not specified	Determinants/characteristics of children employed in hazardous labour (including domestic work)	**CDW:** 1.82% (491,441/26,973,298, estimated, from 1129/60,678 crude figure)**Denominator:** children aged 10–17 in sample	Poor	Poor
5	Levison & Langer 2010 * [20]	CDW in Argentina, Brazil, Chile, Colombia, Costa Rica, Mexico	CDW aged 10–17 N = various	**Measurement tool:** IPUMS-International census microdata samples from 1960 to 2002 **Sampling method:** Census in each country	Number of CDW by country, year, age	CDW weighted estimates vary by country, year, age, live in/out status **Live in CDW:** Brazil (2000) aged 10–14: 11,600, aged 15–17: 46,200Mexico (2000) aged 10–14: 7600, aged 15–17: 44,100 **Denominator:** Various. By % in labour force, age groups, CDW prevalence method type	Poor	Poor
6	Erulkar & Mekbib 2007 [17]	Adolescents in slum areas of Addis Ababa, Ethiopia	Adolescents aged 10–19 (N = 1074, including 99 female CDW)	**Measurement tool:** Structured questionnaire. Population-based household survey (2004) **Sampling method:** Baseline census of all households in the study area to capture basic demographics from all household members (regardless of age). Subsequent random sampling of 1200 households—among households with >1 adolescent (aged 10 to 19), Kish grid used to select 1 adolescent	Prevalence of female adolescent domestic workers, self-esteem, Social connection & support	**Female CDW aged 10–19:** 14.6% (99/676) **Denominator:** females aged 10–19 in sample	Good	Poor
7	Aberra et al. 2003 [15]	Child labourers and non-working controls in Shebe (rural town), Ethiopia	Child labourers aged 5–14 N = 289, including CDW N = 176	**Measurement tool:** Structured questionnaire household survey (2001) **Sampling method:** Sample size derived from estimated total population in Shebe from Municipality office records + arbitrary estimate of child labour (50%). Systematic sampling to recruit study participants: 1st household selected via lottery method; subsequently 1 child from every 4th household was selected (lottery method applied for households with >1 child)	Prevalence of child labour and associated problems (abuse and injury)	**CDW (paid or unpaid):** 77.2% (176/228) **Denominator:** child labourers aged 5–14 in sample	Poor	Moderate
8	Budlender & Bosch/ILO 2002 * [18]	CDW in South Africa	Working children aged 5–17 N = 3,476,358 (including CDW N = 53,942) (no crude N, weighted estimates)	**Measurement tool:** Household survey (SIMPOC SAYP, 1999) **Sampling method:** Phase 1: 30,550 households surveyed in 9 provinces, which provided information on 33,000 children aged 5–17. Second phase: probability sub-sampling for detailed survey on children’s activities of 6110 households containing at least one child doing work of some kind, which collected information on approximately 10,000 children. Results for both phases weighted to make them representative for 5–17 y/os in South Africa	Prevalence of CDW and occupational risks and injuries	**CDW aged 5–17:** weighted estimate 2% (53,942/3,476,358), 62% male **Denominator:** children doing economic work	Poor	Poor
9	NIS/ILO 2004 [19]	CDW in Phnom Penh, Cambodia	CDW aged 7–17, N = 293 in N = 2500 households	**Measurement tool:** Household surveys in Phnom Penh and one migrant sending province + surveys with CDWs, parents/guardians of CDW in origin villages and village chiefs **Sampling method:** Sample frame based on 1998 census. First stage: random selection of 125 villages without replacement. Second stage: Villages with > 200 households selected to divide into clusters, randomly chosen for household listing. Third stage: Linear systematic sampling of 20 households from listing in each of 125 villages. Fourth stage: identify CDWs in survey, then revisit to conduct detailed interview with adults, CDWs and their parents, depending on which situation the household presented	CDW prevalence, violence prevalence, work-related illness and injury, mental health	**CDW aged 7–17:** weighted estimate 9.6% (27,950/292,119) (crude 293/2500 households, all live in), 59% female **Denominator:** total estimated number of children in Phnom Penh	Moderate	Moderate
10	ACPR/ILO Bangladesh 2006 [10]	CDW in Bangladesh	CDW aged 5–17, N = 3841 in N = 3805 employer households	**Measurement tool:** Household survey, + surveys with CDWs and employers in selected households **Sampling method:** Two-stage cluster sampling. 725 urban and rural Primary Sampling Units (PSUs) selected in 5 cities (excluding Dhaka) based on 2001 census data using circular systematic method with probability proportional to size. First stage: after dividing regions into stratum, PSUs divided into equal segments of 125 households, one segment purposively selected where CDW concentration expected to be high, another segment randomly selected, for household listing (N = 167,051). Second stage: sampling frame drawn up of households with CDW. Six households in segments with high CDW concentration and 4 households from other segments then selected via simple random sampling without replacement	CDW prevalence, violence prevalence, work-related illness and treatment seeking	**CDW aged 6–17:** weighted estimate 421,486, 78% female. Overall, 1.1% of all households employ CDWs **Denominator:** All households in Bangladesh	Moderate	Good
11	IER/ILO Vietnam 2006 [16]	CDW in Ho Chi Minh City, Vietnam	CDW aged 6–17, N = 100 Employers N = 10Parents N = 8	**Measurement tool:** structured surveys with CDW, employers and guardians **Sampling method:** Two-stage cluster sampling + snowball sampling. First stage: Random selection of 100/8989 clusters of households in 8 selected districts (divided into core districts, districts with family businesses with CDW, districts based on probability proportional to size) for household listing, to identify CDW. Second stage: Random sampling of CDW listed households in 100 clusters, finding 20 CDW. Repeated for another 100 clusters, findings 19 CDW. Third stage: Snowball sampling of 61 CDWs (most interviewed without employer’s permission).	CDW prevalence & characteristics, work-related abuses and health	**CDW aged 6–17:** weighted estimate 2162 (crude 39/200 households) in Ho Chi Minh City, 70% female **Denominator:** Not specified	Poor	Poor
12	Suhaimi & Farid/ILO 2018 [12]	Domestic workers and CDW in Indonesia	CDW aged 10–17 Domestic workers N = 136 in 1000 households (probing module)	**Measurement tool:** household survey, with an additional module onto the standard Labour Force Survey questions probing domestic work tasks **Sampling method:** Stratified 4 stage sampling. Sample frame based on census (2010). First stage: 10 districts selected as primary sampling units (PSU) by probability proportional to size (PPS) on number of live-in DWs based on census, stratified by typology. Second: 10 clusters (villages) in each district selected by PPS of live-in DWs based on census. Third: Simple random selection of sub-villages to conduct household listing. Fourth: Systematic sampling of 10 households in each selected sub-village	Prevalence of domestic workers, including CDW, based on a survey module used to identify adjustment factors applied to standard LFS data for revised, more realistic DW/CDW estimates	CDW weighted estimates vary by year, example: **CDW aged 10–17:** adjusted & weighted estimate 85,574 (2015), 93% female **CDW aged 10–17 in LFS:** 31,000 (2015) **Denominator:** Not specified Relative Standard Errors larger for CDW than for DWs overall. Higher possibility of under coverage of CDW in Java regencies	Moderate	Moderate
13	Lyon & Valdivia 2010 * [7]	CDW in Paraguay, Uganda, Venezuela	CDW aged 10–17, varies by country and type of CDW	**Measurement tool:** Household surveys in 3 countries with child labour modules, used to estimate CDW prevalence in 3 categories: (1) Commuting and live in CDWs; (2) Live in CDWs only; (3) CDWs under guise of fostering or adoption. Surveys mainly based on World Bank Living Standards Measurement Surveys (LSMS), some based on ILO SIMPOC and UNICEF MICS. **Sampling method:** Not specified	Prevalence of CDWs based on different categories of commuting, live-in and ‘disguised’ (via fostering and adoption) CDWs	CDW estimates vary by category, country, example: **CDW aged 10–17 in Paraguay:** weighted estimate 4.0% (43,792) including all CDW categories **Denominator:** all children aged 10–17	Poor	Poor
14	FAFO 2015 [13]	CDW and working children in Haiti	Children aged 5–17 N = 1617 in N = 2078 households	**Measurement tool:***Haiti Child Domestic Workers Survey* (HCDWS 2014). Household survey + questionnaire for randomly selected child **Sampling method:** Two-stage cluster sampling, with sample frame based on census (2003), stratified by urban/rural. Stage 1: 80 randomly selected clusters based on PPS to the number of households in each cluster. Each cluster was mapped, and households listed and screened for the presence of children not living with parents, with 13,402 households visited for screening. Two lists made in each cluster: one for households hosting children without parents, second for households with children not separated from parents. In each cluster, 20 households in the ‘without parents’ list, 7 households in the ‘with parents’ list, were randomly selected. Households without children aged 5–17 were ineligible	Prevalence of CDW, violence (physical or sexual), physical health problems and depression symptoms	**CDW aged 5–17:** weighted estimate 407,000 (95% CI: 335,000–494,000) (crude 727 in sample) **CDW aged 5–9:** 7% (5.3–9.2) **CDW aged 10–14:** 16.3% (12.5–21.1) **CDW aged 15–17:** 17% (12.4–22.9) **Denominator:** various. By age group and permissible hours (95% CI stated above), living with/without parents, education status	Moderate	Poor

* secondary analyses. ^ assumption.

**Table 2 ijerph-18-02357-t002:** Prevalence measurement methods and question modules in included studies (*n* = 14) #.

#	Study	Household Roster	Occupation Based (ISCO)		Industry Based (ISIC)		Task List Based ***		Unclear/Not Reported	Live in and Live out Estimates ^†^
			HH Head	Direct Report **	HH Head	Direct Report	HH Head	Direct Report		
1	Kedir & Rodgers 2018 *			X						Live in only
2	Gilbert et al. 2018 *			X						Live in only
3	Dalal et al. 2016 *	X								Live in only
4	DeGraff et al. 2016 *			X						Unclear
5	Levison & Langer 2010 *^	X		X		X				Both
6	Erulkar & Mekbib 2007 ^	X		X						Live in only
7	Aberra et al. 2003								X	Unclear
8	Budlender & Bosch ILO 2002 *								X	Unclear
9	NIS/ILO Cambodia 2004	X	X	X	X		X	X		Live in only
10	ACPR/ILO Bangladesh 2006						X	X		Live in only
11	IER/ILO Vietnam 2006								X	Live in only
12	Suhaimi & Farid/ILO 2018	X	X		X		X			Both
13	Lyon & Valvidia 2010 *^	X		X		X				Both
14	FAFO 2015	X	X				X	X		Live in only
TOTAL ^^		7	9		4		4		3	4

* secondary analyses. ** e.g., ‘are you/have you ever been a domestic worker?’ OR ‘what is your main occupation?’. *** only Suhaimi and Farid (2018) uses the task-based module to formally estimate prevalence. ^ includes non-relatives/other relatives in household head relationship. ^^ Total of studies using method. ^†^ some ‘live in only’ studies include survey questions to determine live out prevalence, but do not include these estimates in the report. *# Detailed questions in*
Appendix A.

## Data Availability

Data is contained within the article and Appendix A.

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
