# Peer review of "Suitability of Measurement Tools for Assessing the Prevalence of Child Domestic Work: A Rapid Systematic Review"

_ijerph, 2021, doi:10.3390/ijerph18052357_

Round 1

Reviewer 1 Report

This manuscript gives a meaningful overview of measurement methodology of Child Domestic Work, which is a challenging task due to the variations in definition of the phenomenon of CDW and the invisibility of it in households.

Important point that is conveyed well: ‘fostered’ or adopted children doing unpaid household chores are often not reported as workers (line 57).

The discussion was excellent, especially the section on task-based methods. The findings will be helpful for anyone’s future research on CDW.

A few places need correction as follows:

Your introduction:

The first sentence doesn’t make sense:

“Child domestic work is a largely invisible form of child labour, and an even less visible type of domestic work.”  CDW is for sure an invisible form of child labor, but how can you argue it’s even less visible among domestic work?

The second sentence:

“Child domestic work is a global phenomenon that affects the lives and futures of millions of girls around the world.”: You should rather write “children”, not “girls”, as 1/3 of 17 millions equals millions of boys!

Line 33: Child Domestic Workers (CDW): This acronym should be “CDWs”, not “CDW”.

Fig. 1: First cell: “as” should be “of” of “for”.

Table 1: The acronym PSU should be spelled out, at least the first time.

Line 279: Insert “:” before “Exploring”.

There are a few minor syntax/grammar issues, for instance a few places where there should be a semicolon instead of a comma.

Reviewer 2 Report

This systematic review of studies involving enumeration of child domestic workers across the globe is a competent use of an established methodology. In particular, it usefully summarises some of the difficulties in defining child domestic work for research purposes, it usefully points to the superiority of including a task-based methodology in enumerating child domestic work, and it discusses the different techniques used and problems in developing a unified standard. It is not clear to me, however, that the paper makes a contribution to health outcomes for child domestic workers.

One problem is that it does not discuss the terms “child labour” and the related “child domestic worker”, apparently assuming that the meanings of both terms are clear. Although “child labour” is frequently defined as work that is in some way harmful to children (in which case its relation to public health is not problematic), the term is more frequently used, especially in statistical work, with some kind of standardised, objective, and usually legal definition based on physical age, which does not always coincide with benefits and harm to the children concerned when their well-being is considered holistically: benefits and harm relate to the contexts of the work in children’s lives – material, psychological and social. The dissonance between these two definitions of child labour is likely to become more pronounced in the wake of economic and social disruptions caused by the Covid pandemic. In particular, holistic outcomes of child labour likely to become even more varied when legal definitions are used.

Your paper mentions that outcomes of child domestic work were unclear in all the papers you considered (p.12, l.229; 17, 170); but the qualitative literature shows very varied outcomes, depending on the conditions of work (e.g., Hasketh et al.[1]), the situation of the children concerned, and sometimes on the local culture (e.g., Gamlin et al.[2]). Your paper mentions potential damaging outcomes raised in a number of other studies, without considering the prevalence of such harm in different situations (pp.17-18). Indeed some of these damaging outcomes are all too common in child domestic work and need urgent attention; but similar damaging outcomes can arise from children doing chores within their own elementary families (which you exclude). In some situations children derive real benefits from working in other families (see your reference 21), including improved nutrition and living conditions and sometimes better access to education, all of which you ignore although they can be very relevant to health in situations in which they are sufficiently prevalent. A problem in discussing outcomes is that they are very difficult to assess, especially in survey work – correlations cannot be assumed to indicate causes.

The paper would be strengthened by introducing the reader to the varied situations in which child domestic work occurs, including an indication of reasons why children and their families undertake this kind of work, and the vary varied relations between workers and employers. This kind of information would come as background from qualitative studies rather than the statistical studies the paper is reviewing, and could lead into the kind of information needed to for relevance to health outcomes.

Although your paper notes a need for attention to local cultural contexts, it is not clear that you are fully aware of problems of applying your categories of labour and work to different cultures. Why, for example, in a culture where child rearing routinely involves movement between nuclear families should there be a different classification between children doing chores for their parents and doing chores for relatives or in a fostering situation (p.3, l.100)? If payment for work is a key criterion (p.12, l.274), what about cultures in which children are paid for the chores they do at home? Indeed, why in any study of harmful child domestic work should unpaid harmful work done by children in their own homes be excluded? When we know that the development of competencies in children varies greatly across cultures and expectations (e.g., Rogoff [3]), why should physical age be a factor in the deciding whether or not work relates to health?

To summarise, the paper needs to clarify what kind of prevalence it is concerned about. If it is concerned with the prevalence of work that is harmful to children, considerations of harm and benefits need to come explicitly into the discussion of appropriate criteria – task-related data could be very useful, especially if it includes time spent in work. If, on the other hand, it is concerned only with some classification for the purpose of labour statistics, it is not clear what this has to do with health outcomes.

On a technical point, although I did not systematically check the references, your references 21-23 do not obviously relate to the text where they are cited, which is about studies of violence (P.16, l.112).

Round 2

Reviewer 2 Report

No further comments